# Online Control of Adaptive Large Neighborhood Search Using Deep Reinforcement Learning

**Primary Keywords:** *(2) Learning*

## Abstract

The Adaptive Large Neighborhood Search (ALNS) algorithm has shown considerable success in solving combinatorial optimization problems (COPs). Nonetheless, the performance of ALNS relies on the proper configuration of its selection and acceptance parameters, which is known to be a complex and resource-intensive task. To address this, we introduce a Deep Reinforcement Learning (DRL) based approach called DR-ALNS that selects operators, adjusts parameters, and controls the acceptance criterion throughout the search. The proposed method aims to learn, based on the state of the search, to configure ALNS for the next iteration to yield more effective solutions for the given optimization problem. We evaluate the proposed method on an orienteering problem with stochastic weights and time windows, as presented in an IJCAI competition. The results show that our approach outperforms vanilla ALNS, ALNS tuned with Bayesian optimization, and two state-of-the-art DRL approaches that were the winning methods of the competition, achieving this with significantly fewer training observations. Furthermore, we demonstrate several good properties of the proposed DR-ALNS method: it is easily adapted to solve different routing problems, its learned policies perform consistently well across various instance sizes, and these policies can be directly applied to different problem variants. We will make our implementation code publicly available.

## Introduction

Combinatorial Optimization Problems (COP) involve finding high-quality solutions in a large space of discrete decision variables. Due to their inherent computational complexity, often NP-hard, practical approaches for solving these problems typically rely on handcrafted heuristics. Such heuristics are fast yet lack the guarantee of finding good solutions. A well-known and widely adopted heuristic in this domain is Large Neighborhood Search (LNS) (Shaw 1998). It operates based on the ruin-and-recreate principle, iteratively applying *destroy* and *repair* operators to enhance solutions. The Adaptive Large Neighborhood Search heuristic (ALNS) (Ropke and Pisinger 2006a) extends LNS by incorporating a variety of destroy and repair operators in the search. In ALNS, these operators are assigned a weight that influences their selection during each iteration of the search process. These weights are adjusted based on operator performance. ALNS is commonly configured with an accep-

tance criterion to enhance search performance further. This allows for accepting less promising solutions in the search, aiming to break out of local optima and search potentially more promising areas of the search space. ALNS is a popular metaheuristic for solving large-scale planning and scheduling problems, including various routing problems and job scheduling problems (Mara et al. 2022).

A limitation of ALNS is its reliance on weight-based operator selection. This selection only considers past operator performances, missing potential immediate relationships between the current search dynamics and operator choice. Additionally, the efficacy of operators is influenced by their parameter configurations. Existing configuration methods are mainly based on conventions, experience, ad hoc choices, and experimentation (Mara et al. 2022), which may be sub-optimal as they do not consider the problem-specific characteristics nor incorporate online information about the current search process dynamics. Additionally, experimentation faces limitations as the parameters interact, making grid or random search impractical for computationally expensive real-world problems. To address these issues, various machine learning approaches have been proposed that learn how to destroy or repair a solution (e.g., Hottung and Tierney (2020); Gao et al. (2020)). While these approaches have demonstrated good performance, they are problem-specific, making it difficult to adapt them to other problem variants.

To overcome these challenges, this paper proposes DR-ALNS, which integrates Deep Reinforcement Learning (DRL) into ALNS. This integration aims to learn operator selection and control the operator and acceptance criterion parameters. As such, DR-ALNS learns to respond to changes in the search space by reconfiguring the ALNS online. Unlike existing learning-based LNS and end-to-end DRL approaches, DR-ALNS is designed to be problem-agnostic. It does not rely on the specifics of the underlying problem. More specifically, when being applied to different optimization problems, existing methods may require a problem-specific MDP formulation (states, actions, and reward) for training, while we only need to adapt the action space by indicating the number of chosen operators. This makes our method applicable to other problems with limited changes to the setup.

We assess the effectiveness of our method using the Orienteering Problem with Stochastic Weights and Time Win-

dows (OPSWTW). This problem involves selecting and visiting customers in a specific sequence to maximize rewards within a limited time frame. The OPSWTW assumes stochastic times, making it challenging to solve using conventional solvers and standard search algorithms, such as (A)LNS, due to the need to perform many evaluations. Due to its complexity, this problem was selected for the IJCAI AI4TSP competition (Zhang et al. 2023). We summarize the contributions of our work as follows:

- We introduce a DRL-based method for the online controlling of both operator selection and parameter configuration in ALNS. This approach effectively addresses the limitations of the weight-based operator selection procedure and the dependency of operator performances on parameter configurations and greatly enhances ALNS performance, achieving better solutions with much fewer search iterations.
- Our method was evaluated using the instances from the IJCAI AI4TSP competition, where it outperformed four benchmark methods, including two DRL-based competition-winning methods.
- We evaluate the generalizability of the policy learned by DR-ALNS, demonstrating its ability to perform effectively on larger problem instances not encountered during training when only exposed to smaller instances during the learning process.
- We also show that DR-ALNS can effectively be applied to various other routing problems, e.g., CVRP, TSP, and mTSP, by only configuring the actions space with the selected destroy and repair operators. Moreover, the policy learned by one problem can be used directly to guide ALNS for solving the other two problems with good performance without retraining.

## Related Work

Many ML-based approaches have been used to define end-to-end solutions for routing problems (e.g., Kool, Van Hoof, and Welling (2018); Joe and Lau (2020); da Costa et al. (2021)). In parallel, a growing focus is on enhancing iterative search algorithms with ML techniques. Examples include the integration of neural networks with attention mechanisms as repair operators within the LNS framework to solve routing problems (Hottung and Tierney 2020) and the use of a neural network with domain-specific features for solution repair (Syed et al. 2019). Further advancements include (Sonnerat et al. 2021), which employs neural networks in node removal in a routing problem, and Gao et al. (2020), incorporating a graph attention network with edge embedding as an encoder to capture the effect of graph topology on the solution. Further, Wu et al. (2021) uses DRL to select the variables to be removed. These approaches operate end-to-end, approximating a mapping function from the input to the solution, and obtain very good performance on specific problems. However, they often face scalability issues with more complex problem variants. To train these methods effectively, significantly more samples and training times are required. Additionally, these existing end-to-end approaches are designed to be tailored to specific problems and lack the flexibility for adoption to other problems. In contrast, our proposed DR-ALNS method is a hybrid approach that does not attempt to learn to construct solutions directly. Consequently, our method relies significantly less on complex architectures for learning instance representations, resulting in faster training and better generalization capabilities.

Recent works have sought to address these limitations using DRL to select predefined operators based on the state of the search (Kallestad et al. 2023; Johnn et al. 2023). However, Johnn et al. (2023) uses a graph to represent routing problems, limiting its applicability to other types of problems. Similarly, Kallestad et al. (2023) relies on absolute objective values in its state space, limiting its direct application to scenarios not included in the training. Besides, the use of computationally expensive operators (e.g., beam search) makes the approach less effective, particularly when addressing computationally demanding problems like ours. We will further highlight this limitation in our experiments. We propose a new DRL-based approach for both the selection of operators and the parameter configuration within the ALNS algorithm. Compared to the above methods, our DRL modelling is problem-agnostic. Hence, the approach can solve broader COPs, with fairly good performance using only inexpensive operators.

## Deep Reinforced Adaptive Large Neighborhood Search (DR-ALNS)

We illustrate the proposed Deep Reinforced Adaptive Large Neighborhood Search (DR-ALNS) method in Figure 1 and Algorithm 1. DR-ALNS leverages DRL to select the most effective destroy and repair operators, configures the destroy severity parameter imposed on the destroy operators and sets the acceptance criterion value within ALNS. This selection and configuration process is performed online, with DRL configuring ALNS at each search iteration. Unlike other DRL-based approaches that are often tailored to specific optimization problems, our goal is to utilize DRL in a generalizable way such that the MDP formulation and training of our DRL agent do not rely on any information of the optimization problem instances. Below, we explain the components of DR-ALNS.

### General ALNS Algorithm

The Large Neighborhood Search (LNS) algorithm aims to improve solutions by iteratively extracting parts of a solution and relocating them to more advantageous positions using a 'destroy' and 'repair' operator. The destroy operator removes parts of a solution $x$, while the repair operator restores it to produce the next solution $x_t$. Typically, only new solutions that enhance the cost function are accepted. Some studies employ acceptance criteria, such as Simulated Annealing (SA), where a solution $x_t$ is accepted with a probability of $e^{-(c(x_t)-c(x))/T}$ when $c(x) \leq c(x_t)$ (Schrimpf et al. 2000). Here, $T$ represents a temperature that decreases over iterations, permitting more deteriorating solutions at the beginning of the search. ALNS employs a set of destroy operators $d \in \Omega^-$ and repair operators $r \in \Omega^+$ during search (Ropke and Pisinger 2006a). Each operator

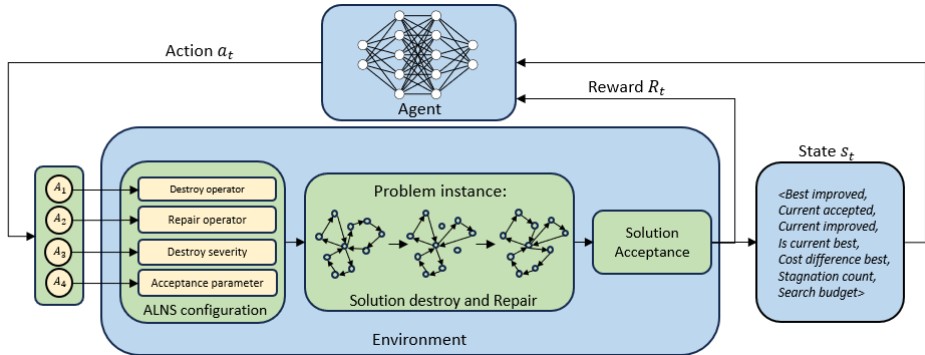

Figure 1: The DR-ALNS framework. Based on the search status in each iteration, the DRL agent chooses a destroy and repair operator from the predefined candidates, determines the level of destruction, and adjusts the acceptance criterion parameter (i.e., simulated annealing temperature). These actions are performed in the environment (i.e., ALNS algorithm), which finds a new solution and returns the next state and reward to the agent

.

is assigned a weight, $\rho^-$ or $\rho^+$, dictating how frequently it is used in the search. These weights are initialized with the same values and updated after each search iteration based on the solution quality as follows: $\rho_i = \lambda \rho_i + (1-\lambda)\psi$ for both $i \in \Omega^-$ and $i \in \Omega^+$, where $\lambda$ is the decay factor controlling weight change sensitivity, and $\psi$ is a parameter score based on the acquired solution quality. The probability of selecting a given operator $i$, either destroy or repair, is computed as: $\phi_i^- = \frac{\rho_i^-}{\sum_{k=1}^{|\Omega^-|} \rho_k^-}$ for $i \in \Omega^-$ and similarly for $i \in \Omega^+$.

**Limitation of traditional ALNS configuration.** The initial proposed ALNS from Ropke and Pisinger (2006a) incorporates SA as its acceptance criterion. Generally, SA requires setting various parameters before deployment, such as the starting temperature $T$, the cooling rate $\alpha$, and parameters of the stopping criterion.Furthermore, the integration of several destroy and repair operators, along with the need to define weights for adaptive operator selection, introduces additional parameters that must be set. Consequently, ALNS parameters are required to be effectively set by practitioners, which is known to be a complex and resource-intensive task. Designers often rely on values from previous literature, but the optimal parameter set is highly problem-specific.

Nevertheless, several parameter tuning methods exist, such as racing-based methods and Bayesian optimization (e.g., (López-Ibáñez et al. 2016; Lindauer et al. 2022)). This tuning typically involves multiple algorithm runs on different problem instances, demanding substantial resources. Besides, these methods suggest one static parameter configuration that does not adapt to the dynamics in the search.is not tailored to the specifics of individual problem instances, configuring each deployment with the same configuration. Moreover, the weights in the weight-based operator selection method do not incorporate the dynamic nature of the search process.

## Markov Decision Process Formulation

We model the ALNS configuration as a sequential decision-making process where the agent interacts with the environ-

---

**Algorithm 1: DR-ALNS**

**Input:** $M$ (number of steps), policy $\pi_\theta$, $\Omega^-$ destroy operators, $\Omega^+$ repair operators, problem instance
**Initialize:** $x_{\text{best}} = $ initial solution $x$,
$s_t = $ initial state
**while** *Stopping criterion not met* **do**
    **for** $t = 0$ *to* $M - 1$ **do**
        Select action $a_t$ with policy $\pi_\theta$ based on state $s_t$
        Select destroy operator $d$ and repair operator $r$
          based on action $a_t$
        Configure destroy severity and acceptance
          criterion based on action $a_t$
        $x_t = r(d(x))$
        **if** $accept(x_t, x)$ **then**
          $x = x_t$
        **if** $cost(x_t) < cost(x_{best})$ **then**
          $x_{\text{best}} = x_t$
        Update state $s_t$ and receive reward $R_t$
    Update policy $\pi_\theta$ (for training mode)
**return** $x_{\text{best}}$ (for deployment mode)

---

ment by taking actions and observing the consequences. This is modelled using a mathematical framework known as a Markov Decision Process (MDP), which is represented as a tuple $\langle S, A, R, P \rangle$. Here, $S$ denotes the set of states, $A$ represents the set of actions, $R$ is the reward function, and $P$ is the state transition probability function. The state transition occurs after the agent executes an action, which leads the environment from the state $s_t$ to state $s_{t+1}$. The agent then receives a reward according to the reward function $R$. The goal of the DRL agent is to learn a policy function $\pi_\theta$ that maps states $s_t$ to actions $a_t$ to maximize the expected sum of future rewards. We define the state space $S$, action space $A$, and reward function $R$ as follows:

**State space** $S$ provides a DRL agent with the required information for making informed decisions for selecting the best possible actions during a search iteration. To achieve this, we formulate $S$ as a one-dimensional vector containing seven problem-agnostic features, as shown in Table 1. These

Table 1: State space features for DR-ALNS.

| Feature | Description |
|---------|-------------|
| Best_improved | Binary feature indicating whether current solution has improved compared to previous iteration (0 or 1). |
| Current_accepted | Binary feature indicating whether current solution has been accepted in the search (0 or 1). |
| Current_improved | Binary feature indicating whether current solution was accepted and is better than previous solution (0 or 1). |
| Is_current_best | Binary feature indicating whether current solution is equal to the best-found solution (0 or 1). |
| Cost_difference_best | Percentage difference between the objective values of the current and best solutions ($-1$ if the current objective value is less than or equal to 0). |
| Stagnation_count | Number of consecutive iterations without improving the best-found solution. |
| Search_budget | Percentage of used search budget. |

features provide the agent with relevant information about the search process, such as whether the current solution is the best solution found thus far, whether the best solution has recently been improved, whether the current solution has been recently accepted and whether the new current solution is the new best-found solution. Additionally, the percentage cost difference from the best solution, the number of iterations without improving the best-found solution, and the remaining search budget in percentages are included.

**Action space** $A$ is composed of four action spaces: *($A_1$) destroy operator selection*, which consists of a set of indices, mapping to the set of destroy operators configured for the problem at hand. The agent selects one destroy operator to apply to the current solution; *($A_2$) repair operator selection* comprises the set of repair operators configured for the problem at hand. The agent selects a repair operator to apply to a destroyed solution; *($A_3$) destroy severity configuration*, which sets the severity of the destroy operator to apply in the next iteration. A higher severity implies that the destroy operator will destroy a larger part of the solution. We configure the destroy severity as discrete actions, ranging from 1 to 10, representing 10% to 100% destroy severity of a solution; and *($A_4$) acceptance criterion parameter setting*, which determines whether a new solution generated by the search procedure is accepted or rejected. We define this action as the temperature $T$ used by the Simulated Annealing acceptance criterion. We set this action space as a discrete set ranging from 1 to 50, where 1 represents $T = 0.1$, and 50 represents $T = 5.0$.

**Reward function** $R$ is formulated for learning to select actions based on the state $S$ of the search process. We provide a reward of 5 when a new found solution $x_t$ is better than the best-known solution $x_{\text{best}}$:

$$R_t = \begin{cases} 5, & \text{if } c\left(x_t\right) > c\left(x_{\text{best}}\right) \\ 0, & \text{otherwise} \end{cases}$$

This score was chosen because it reflects the scoring function proposed in the original ALNS work.

**State Transition Function** $P$ is learned by the agent through interacting with the environment. By formulating the MDP in this way, we provide a problem-agnostic environment for training the agent, making $S$ and $R$ independent from problem specifics. To use DR-ALNS, one only needs to define the destroy and repair operators and create the action space $A$ accordingly.

### Neural Network and Policy Optimization

We utilize the Proximal Policy Optimization (PPO) algorithm (Schulman et al. 2017) to train DR-ALNS. PPO is a widely used and highly effective policy gradient algorithm that maximizes the improvement of the current policy. PPO uses two loss functions, a policy loss and a value loss, where the policy loss measures the differences between the new and the old policy, and the value loss function quantifies the error between the predicted state values and the actual discounted sum of reward. The policy loss is given by the PPO objective function: $L^{CLIP}(\theta) = \mathbb{E}_t[\min(r_t(\theta)\hat{A}_t, \text{clip}(r_t(\theta), 1 - \epsilon, 1 + \epsilon)\hat{A}_t)]$, where $r_t(\theta)$ is the probability ratio of taking actions under the new policy compared to the old policy, and $\hat{A}_t$ is the estimated advantage of taking action $a_t$ at time step $t$. The value loss is given by the squared error between the predicted value and the actual discounted sum of rewards obtained from the current state: $L^{VF}(\theta) = \mathbb{E}t[(V\theta(s_t) - G_t)^2]$, where $V_\theta(s_t)$ is the predicted value of the current state, and $G_t$ is the actual discounted sum of rewards obtained from the current state. The network used for training with the PPO algorithm is an MLP with two hidden layers of size 64. To accommodate multiple discrete actions, the policy network has individual fully connected output layers for each discrete action, each generating a probability distribution via softmax.

## DR-ALNS for solving OPSWTW

We show how to apply the proposed DR-ALNS to solve the Orienteering Problem with Stochastic Weights and Time Windows (OPSWTW). This problem, first introduced by (Verbeeck, Vansteenwegen, and Aghezzaf 2016) and used in the IJCAI AI4TSP competition (Zhang et al. 2023), poses several challenges, such as unknown travel costs between locations, limited travel time, and time windows for customer visitation. In OPSWTW, each customer is represented as a node with a designated prize and time window for visitation, and the objective is to maximize expected collected prizes while respecting the time budget and time windows.

Formally, the problem consists of $n$ customers, each located at coordinates $x$ and $y$. The stochastic travel times $t_{i,j} \in \mathbb{R}, \forall i, j \in \{1, \ldots, n\}$ between customers $i$ and $j$ are computed by multiplying the Euclidean distance $d_{i,j}$ by a noise term $\eta$ that follows a discrete uniform distribution $\mathcal{U}\{1, 100\}$ normalized by a scaling factor $\beta = 100$. Each customer has a time window and a prize that can be collected when visited within the time window. The maximum tour length is determined by $L$, and solutions must respect the time windows and the maximum tour time. Violations of

these constraints incur penalties. Solutions that take longer than $L$ are penalized with $e_i = -n$, and time window violations incur a penalty of $e_i = -1$ at customer $i$.

**ALNS Configuration.** We use three destroy operators from Ropke and Pisinger (2006b): (1) *random removal* uniformly removes $n$ customers from a solution; (2) *relatedness removal* removes customers based on how close visited customers are located to each other. To invoke it, a visited customer is randomly selected and removed, after which the closest customer is iteratively removed from the solution until $n$ customers are removed; and (3) *history-based removal* uses historical information, which is stored in a complete, directed, weighted graph, called the neighbour graph, where each node is a customer and the weight of edges between nodes stores the cost of the best solution encountered so far in which the edge is traversed. When this removal operator is invoked, it computes scores for each customer in the current solution by summing the edge weights in the neighbour graph corresponding to the current solution. The customers with low scores are removed.

Three repair operators were selected based on ALNS applications to orienteering problems, e.g., in Hammami, Rekik, and Coelho (2020); Yahiaoui, Moukrim, and Serairi (2019); Roozbeh, Hearne, and Pahlevani (2020): *random distance repair*, *random prize repair* and *random ratio repair*. In random distance repair, randomly selected customers are inserted into the solution at their least expensive position in terms of distance. Random prize and ratio repair prioritize positions that maximize total rewards or optimize the reward-to-distance ratio. All destroy and repair operators are compatible, allowing any repair operator to fix solutions altered by any destroy operator.

We use roulette-wheel selection to select the destroy and repair operators at each iteration. In this selection mechanism, the probabilities of the operators are proportional to their scores, which are initially assigned the same value. As an acceptance criterion, we use Simulated Annealing (SA), the most frequently used acceptance criterion for ALNS (Santini, Ropke, and Hvattum 2018). We set the cooling schedule of ALNS such that the temperature decays linearly to an end temperature of 0, reducing the number of parameters to tune by one without sacrificing the solution quality (Santini, Ropke, and Hvattum 2018).

To address the stochastic weights of the problem, we evaluate any solution encountered during the repair operator five times and take the average quality. This is commonly done for search-based algorithms in solving stochastic problems. To ensure that feasible solutions are accepted in the search, the 'repaired' solution is evaluated 100 times.

**DRL agent.** To apply DR-ALNS to the OPSWTW problem, we need to configure the action space to incorporate the selected destroy and repair operators. This entails adjusting the dimensions of $A_1$ and $A_2$ in the action space, as displayed in Figure 1, to align with the selected destroy and repair operators. Specifically, for OPSWTW, where three destroy and three repair operators are utilized in ALNS, we create two three-dimensional vectors: $< d_1, d_2, d_3 >$ for destroy operators and $< r_1, r_2, r_3 >$ for repair operators. The

DRL policy will learn to select one operator from each vector that will be subsequently applied in the next iteration of the ALNS search. The remainder of the action space, the state space and the reward function remain in line with the description provided in the previous section.

## Experiments

We use publicly available OPSWTW instances from the competition[1], which consist of 3 sets of 250 instances each, with 20, 50, and 100 customers. We compare the proposed DR-ALNS against four methods: a vanilla ALNS algorithm (ALNS-Vanilla), an ALNS tuned with Bayesian optimization (ALNS-BO), and the top two winning DRL-based methods of the competition, Rise_up and Ratel. We also compare with DRLH (Kallestad et al. 2023) briefly. We measure these methods regarding solution quality (i.e., total collected prize - penalties), training time, and inference time.

**ALNS-Vanilla.** To configure ALNS-Vanilla, we adopt operator weights from (Ropke and Pisinger 2006a), setting weights $\omega_1, \omega_2, \omega_3$, and $\omega_4$ to 5, 3, 1, 0, respectively. Specifically, a score of 5 is given to the operator when a new best solution is found, 3 when the current solution is improved, and 1 for accepting a solution. No score is given when there is no improvement or accepted solution. The initial temperature $T_{start}$ for SA follows a rule-of-thumb of Roozbeh, Ozlen, and Hearne (2018), where in the first iteration, a solution with an objective value up to 5% worse than the initial solution is accepted with a probability of 0.5. The initial solution quality is determined using the random prize repair operator. In addition, we set the degree of destruction $dod$ at 30% and the decay factor $\theta$ as 0.8 based on experimental trials. This degree of destruction controls how much of a solution is removed by the destroy operator, and the decay factor controls the rate at which the weights of the operators are updated during the search process.

**ALNS-BO.** We used SMAC3 to tune the parameters of our ALNS algorithm (Lindauer et al. 2022). SMAC3 is a hyperparameter tuning method that combines Bayesian optimization and random forest regression. To do this, we generated 25 instances for each given instance size and used them solely for the tuning. We tuned the weight factors $\omega_i$, the decay parameter $\theta$, the degree of destruction $dod$, and the Simulated Annealing starting temperature $T_{start}$. The configuration ranges we considered were $[0, 50]$ for each $\omega_i$, $[0.5, 1]$ for $\theta$, 10% to 100% for $dod$ and $[0, 5]$ for $T_{start}$. We set $\omega_4$ at 0. Bayesian optimization was then used to draw parameter configurations from these ranges and evaluate them on the provided tuning instances over 100 ALNS search iterations. We performed 25 independent runs, lasting 12 hours for the two smallest-sized instance sizes (20 and 50 customers, respectively) and 24 hours for instances of 100 customers. The best configurations in these independent runs are presented in Table 2 and are used as a baseline. The tuning results reveal that the obtained weight factor ratios are comparable to those in (Ropke and Pisinger 2006a). Moreover, the initial temperature $T_{start}$ rises as the instance sizes grow. This

---

[1]https://github.com/paulorocosta/ai-for-tsp-competition

Table 2: Tuned ALNS-BO parameters for different instance sizes of the OPSWTW problem.

| Instance size | $\omega_1$ | $\omega_2$ | $\omega_3$ | $T_{start}$ | $dod$ | $\theta$ |
|---|---|---|---|---|---|---|
| 20 | 28.2 | 22.6 | 9.9 | 1.2 | 0.45 | 0.6 |
| 50 | 21.5 | 6 | 1.65 | 1.35 | 0.27 | 0.75 |
| 100 | 34.3 | 27.5 | 17.6 | 1.37 | 0.35 | 0.65 |

trend aligns with the rule-of-thumb proposed by Roozbeh, Ozlen, and Hearne (2018), which sets the initial temperature based on the initial solution quality, which is generally larger for larger OPSWTW problem instances.

**Rise_up and Ratel.** *Rise_up* (Schmitt-Ulms et al. 2023) is a combination of an end-to-end DRL method named POMO (Kwon et al. 2020), Efficient Active Search (EAS) and Monte-Carlo roll-outs (MCR). POMO trains a DRL agent to construct solutions directly, using an encoder and a decoder neural network that exploits symmetries to encourage exploration during learning. For solving OPSWTW, the method provides problem-specific information to the input for each node, including instance prizes, time constraint information, travel times, and the embedding of the current node in a route and of the depot. Masking is used to prevent invalid and infeasible actions from being taken. The EAS procedure was used to fine-tune the learned policies for each instance. Finally, MCR are used to construct the final solutions by sampling actions with the learned policies.

*Ratel* is an end-to-end DRL approach, adapted by the authors of Gama and Fernandes (2021) for the IJCAI competition, using a Pointer Network (PN) to handle problems with dynamic time-dependent constraints. The model contains a set encoding block, a sequence encoding block, and a pointing mechanism block, with recurrence in the node encoding step. This recurrence enables sequential encoding and decoding steps for every step of the solution construction process, allowing masked self-attention with a look-ahead-induced graph structure. This results in an updated representation of each admissible node in every step. The method uses a feature vector associated with every admissible node as input, combining 11 static and 34 dynamic features.

**DRLH.** DRLH (Kallestad et al. 2023) provides a DRL framework. Despite some similarities to our method, the DRLH employs several computationally expensive operators, each configured with predefined destroy degrees, and a fixed Simulated Annealing cooling schedule. This makes the method less flexible to other COPs and potentially less effective than ours. To demonstrate this, we implemented its beam search repair operator and applied it to OPSWTW instances with 20 and 50 customers, running for 100 iterations. The resulting inference times were 300 and 10k seconds, as opposed to 4 and 12 seconds achieved by our ALNS method. This underscores lightweight nature of the operators presented in our study. We correspondingly trained DRLH for 5 hours (10x more than DR-ALNS) on instances of size 20. The results were, on average, 1.0% (gap) worse than ours on the test instances. Given the extreme increase in inference times, we were unable to train any effective policy for

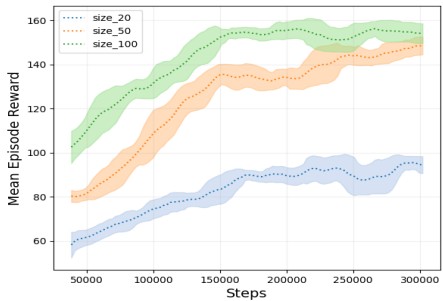

Figure 2: Rolling mean and standard deviation of training episode rewards over time for varying instance sizes.

DRLH on larger instances. Hence, we will not further compare DRLH with our method.

## Model Training for DR-ALNS

We trained three models with randomly generated problem-instance sizes of 20, 50, and 100 customers, each trained with 250 different problem instances. The training process involved 300,000 steps with 100 search iterations. We conducted the training on a Processor Intel(R) Core(TM) i7-6920HQ CPU @ 2.90GHz with 8.0GB of RAM and ten parallel environments. The training duration varied for different-sized instance sets, taking around 0.5, 2.5 and 10 hours. The model parameters are set following Schulman et al. (2017), and the training traces are displayed in Figure 2, showing the mean reward during training. The figure shows that the training runs follow a similar convergence pattern. They demonstrate a quick improvement in gathering more rewards early in the training process, followed by convergence to a state where they consistently earn high rewards. Notably, the models can better obtain more rewards per episode for larger instances. This can be attributed to larger problem instances presenting a greater variety of routing options to explore during the search process.

## Experimental Results

We evaluate the performance of different algorithms on instances of the OPSWTW problem (Zhang et al. 2023). ALNS-Vanilla, ALNS-BO and DR-ALNS are initialized with an empty route, with a solution quality of 0.00. The stopping criteria for the ALNS-based approaches are set to 100 iterations for the sets with 20 and 50 customers and to 200 for 100 customers. Each method is run 50 times with different random seeds. The best-found solutions of each method are correspondingly evaluated with the solution evaluator of the competition, in which the seed has been fixed to ensure a fair comparison. The results from the Rise_up and Ratel approaches are obtained by evaluating their submitted solutions to the AI4TSP competition.

**Solution quality.** In Table 3, the performances of different methods are presented, including the average performance of the best solution obtained for each method on each problem instance size and the number of best-found solutions. The results indicate that the DR-ALNS method outperforms

Table 3: Performance comparison of different methods in solving 250 instances of varying sizes, based on average best solutions found (Avg) and the number of times the best solution was found (Nr. Best) per method. DR-ALNS significantly outperforms the other methods ($p < 0.05$) for all instance sizes, except for the Ratel method on instance sizes 100.

| | Rise_up | | Ratel | | ALNS-Vanilla | | ALNS-BO | | DR-ALNS | |
|---|---|---|---|---|---|---|---|---|---|---|
| Instance Size | Avg | Nr. Best | Avg | Nr. Best | Avg | Nr. Best | Avg | Nr. Best | Avg | Nr. Best |
| 20 | 5.47 | 159 | 5.53 | 183 | 5.52 | 185 | 5.55 | 195 | **5.63** | **238** |
| 50 | 8.27 | 131 | 8.31 | 145 | 8.27 | 126 | 8.23 | 109 | **8.44** | **208** |
| 100 | 11.67 | 131 | 11.71 | 148 | 11.26 | 54 | 11.36 | 66 | **11.75** | **180** |

Table 4: Performance comparison of differently configured ablations of the DR-ALNS method, configured with the control of operator selection (OS), operator selection and the destroy severity parameter (OS+D) and operator selection with acceptance criterion parameter (OS+A). Performance measured on average best solution found for instances of size 50.

| ALNS-BO | DR-ALNS (OS) | DR-ALNS (OS+D) | DR-ALNS (OS+ACC) | DR-ALNS (OS+D+ACC) |
|---|---|---|---|---|
| 8.23 | 8.28 | 8.31 | 8.31 | **8.44** |

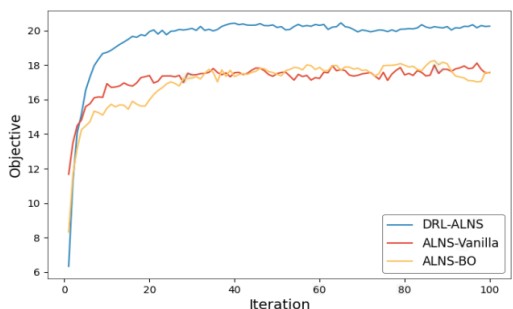

Figure 3: Average convergence comparison for ALNS-Vanilla, ALNS-BO, and DR-ALNS on a 100-node problem.

all other benchmark methods in terms of the best average solutions, and it finds the best solutions for instances more often than the other methods. The table demonstrates the effectiveness of our approach in controlling the parameters of ALNS. For the smallest problem instance size, the ALNS-Vanilla and ALNS-BO perform competitively, but their performances decline as instance sizes become larger. Bayesian optimization effectively tunes the ALNS, producing better solutions than the vanilla-configured ALNS for instances of size 20 and 100. In Figure 3, we show that DR-ALNS can find the best solution in much fewer iterations than ALNS-Vanilla and ALNS-BO for one specific problem instance. This convergence pattern can also be found in other problem instances.

Compared to the two DRL-based approaches, DR-ALNS consistently performs better for different instance sizes. Note that Rise_up and Ratel rely on information specific to problem instances for learning, whereas our approach is problem-agnostic, making it potentially applicable to other problem types without hands-crafted feature engineering.

**Ablation Study.** To gain a deeper understanding of the contributions made by the various components controlled by DRL within the DR-ALNS method, we undertook an ablation study. As such, we configured ablations where DRL controlled only operator selection (OS) and tested ablations where DRL also controlled either the destroy sever-

ity parameter (OS+D) or the acceptance criterion parameter (OS+A). Table 4 compares the average best solutions for solving instances of size 50 by the different ablated variants.

The table shows that learning to select operators effectively improves the ability to find good solutions, compared to the Bayesian-tuned ALNS (ALNS-BO). Also, we observe that both the destroy severity and acceptance parameter setting components contribute to even better-found solutions. From the results can also be concluded that the different components complement each other, highlighted by the fact that the DR-ALNS method acquires substantially better performances than the different ablations.

**Computation time.** We compare the computation time of various methods, i.e., the time for obtaining individual solutions for instances of different sizes. For instance sizes of 20, the Rise_up method took 7 minutes, while ALNS-Vanilla and ALNS-BO needed just 4 seconds, and DR-ALNS 5 seconds. For the 50-sized instances, the time for Rise_up, ALNS-Vanilla/ALNS-BO, and DR-ALNS is 13 minutes, 12 seconds, and 30 seconds, respectively, and for instance sizes of 100, 23 minutes, 2 minutes, and 3.5 minutes, respectively. This highlights that the proposed DR-ALNS method requires substantially less computational time than the Rise_up method while finding better solutions for the different-sized problem instances. Even though DR-ALNS adds some overhead in solving time compared to ALNS-Vanilla and ALNS-BO, we have shown in Figure 3 that DR-ALNS can find better solutions much faster. As Ratel is composed end-to-end, it can generate solutions in a split second. Despite its lower performance, it is proficient at generating reasonably effective solutions in a short time.

Regarding training time, we trained DR-ALNS for 0.5, 2.5, and 10 hours for instances of sizes 20, 50 and 100, respectively. The Rise_up is configured with a distinct policy model for each problem size. These models were trained for several days until they reached full convergence on a single Tesla V100 GPU, after which they were tuned for individual instances. The Ratel method was trained for 48 hours on a 6-core CPU at 1.7 GHz with 12 GB of RAM and an Nvidia GeForce GTX 1080 Ti GPU (Zhang et al. 2023).

Table 5: Comparing the generalizability of the trained models to solve unseen instances of different sizes.

| Instance Sizes | 20 | 50 | 100 |
|---|---|---|---|
| DR-ALNS-20 | 5.63 | 8.35 | 11.58 |
| DR-ALNS-50 | **5.65** | 8.44 | 11.63 |
| DR-ALNS-100 | 5.64 | **8.48** | **11.75** |

**Policy Scalability.** We assess the ability of the trained DR-ALNS models to solve previously unseen instances of different sizes. The results of this evaluation are presented in Table 5. The rows present the instance sizes on which the model is trained, and the column shows the instance sizes on which trained models are evaluated. We found that the model trained on smaller instances and deployed on larger instances experienced a slight decline in performance but still managed to find better solutions compared to all the benchmarks provided. Moreover, the models trained on larger instances and deployed on smaller instances could improve results further. For instance, models trained on instances with 50 and 100 customers and deployed on instances with 20 and 50 customers found better solutions than models directly trained on these smaller-sized instances. The results suggest that our models could generalize and solve problem instances beyond the size on which they were trained.

## DR-ALNS for solving other routing problems

DR-ALNS is problem-agnostic, making it potentially applicable to control ALNS for other COPs with minimal setup modifications. We demonstrate this generalizability by applying it to the ALNS configuration from Santini, Ropke, and Hvattum (2018) for solving several routing problems: Capacitated Vehicle Routing Problem (CVRP), Traveling Salesman Problem (TSP) and multi Traveling Salesman Problem (mTSP) . Three destroy operators and one repair operator are utilised by ALNS. Accordingly, $A_2$ in the action space (see Figure 1) becomes one dimensional. All remaining MDP formulations stay the same as those for OPSWTW.

Table 6 shows the average distance for 5,000 instances with 100 nodes for each problem. Instances are obtained according to (Kool, Van Hoof, and Welling 2018; da Costa et al. 2021) and solved once with both methods with 1k and 10k search iterations. The table shows that DR-ALNS obtains solutions that outperform those from the ALNS. This is especially highlighted in its performance on CVRP, the most challenging routing variant among the three problems. Here, we observe that with effective control by the DRL policy, better solutions can be obtained with 10x fewer search iterations. This is also shown in Figure 4.

**Transferability across problems.** We utilize a general ALNS configuration for routing problems, allowing us to train on one routing problem and apply the learned policy directly to guide ALNS in solving other similar problems. Table 7 shows the relative difference between the performance of models trained on one problem directly and the one trained on different problems. We notice the model trained on mTSP can be transferred effectively to TSP, obtaining similar performances and performing reasonably well on

Table 6: Comparing ALNS-Vanilla and DR-ALNS on 5,000 CVRP, TSP, and mTSP instances with 100 customers. Values are average travel distances (lower, better).

| | ALNS-Vanilla (1K) | DR-ALNS (1K) | ALNS-Vanilla (10K) | DR-ALNS (10K) |
|---|---|---|---|---|
| CVRP | 17.84 | **16.80** | 17.19 | **16.38** |
| TSP | 7.86 | **7.82** | 7.79 | **7.78** |
| mTSP | 8.52 | **8.50** | 8.41 | **8.39** |

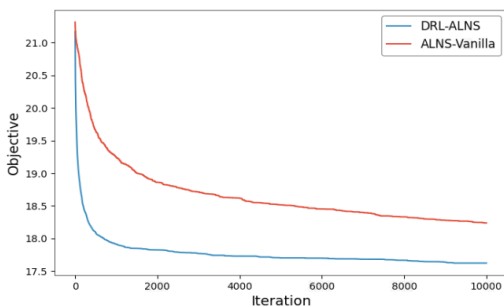

Figure 4: Average convergence comparison on a CVRP problem instance with 100 customers.

Table 7: Performance comparison of direct model applications across different problems.

| | TSP | CVRP | mTSP |
|---|---|---|---|
| Trained TSP model | 0% | 6.8% | 0.4% |
| Trained CVRP model | 1.2% | 0% | 5.2% |
| Trained mTSP model | 0% | 3.3% | 0% |

CVRP, outperforming ALNS-vanilla. Further, TSP can be transferred to mTSP, and the model trained on CVRP performs reasonably on other problems.

## Conclusion

We propose DR-ALNS, a Deep Reinforcement Learning-based approach for online controlling and configuring the Adaptive Large Neighborhood Search heuristic. Our method selects operators, configures the destroy severity parameter and controls the acceptance criterion via the simulated annealing temperature during the search. We demonstrate the effectiveness of our approach on the complex orienteering problem with stochastic weights and time windows. Our method outperforms vanilla ALNS, ALNS tuned with Bayesian optimization, and obtains better solutions than two state-of-the-art DRL approaches. Furthermore, we show that the proposed approach can easily be adapted to other problems. Specifically, the trained policy from one routing problem can effectively guide ALNS in solving other routing problems and scales to larger problem instances that were not included in the training. In the future, we plan to investigate the performance of the framework on other combinatorial optimization problems.

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
