# OpenReview forum: "Online Control of Adaptive Large Neighborhood Search Using Deep Reinforcement Learning"
_icaps-conference.org/ICAPS/2024/Conference — ICAPS 2024_

### Official Review · Reviewer_L86V · 2024-01-15

**Significance And Importance:** 2
**Soundness:** 3
**Novelty:** 2
**Clarity:** 3
**Overall Evaluation:** 1
**Confidence:** 4

**Weaknesses:**

1: Minor weaknesses that are easily fixable.

**Contributions Of The Paper:**

The paper proposes a deep reinforcement learning (RL) approach, called DR-ALNS, to enhance LNS with adaptive operator, parameter, and acceptance criterion selection. DR-ALNS only depends on very limited features, thus being problem-agnostic in contrast to alternative RL approaches which are specifically designed for particular problem classes. The approach is extensively evaluated against vanilla LNS and alternative RL variants in instances of the IJCAI AI4TSP competition. The paper further evaluates the generalization capabilities to larger problem instances and applicability to alternative routing problems.

**Ethical Considerations:**

(1) Not Applicable: The paper does not have any ethical considerations to address

**Nomination For Best Paper:**

No

**Questions For Authors:**

Why is the numerical reward 5 for an improvement? Wouldn’t just 1 be sufficient?

**Reproducibility:**

3: Authors describe the implementation and domains in sufficient detail.

**Strengths Of The Paper:**

The paper is well-written and easy to understand.

The proposed approach does not use end-to-end learning but is integrated into the search procedure, which simplifies the architecture and learning methods.

I like the extensive evaluation that covers all necessary aspects in my view.

**Weaknesses Of The Paper:**

From an algorithmic perspective, the proposed approach merely applies PPO (a well-known RL algorithm) to ALNS without further innovation.

I am concerned about the soundness of the MDP formulation: The information contained in the features specified in Table 1 seems very limited as they do not contain any characteristics of the actual solution. While this makes the approach indeed problem-agnostic, the Markov property is likely to be violated as it requires “full knowledge” about the underlying state (which is not provided here). It may be valid for a POMDP formulation, though.

Minor remarks:
- Given that the state space has 7 features, it is 7-dimensional rather than “one-dimensional” as specified in the text
- “State Transition Function P is learned …” - PPO is a model-free algorithm, thus does not learn any environment dynamics (it is implicitly considered in the value function but not learned explicitly).

---

> ### Author Rebuttal · Authors · 2024-01-26
>
> **MDP formulation**
>
> With our MDP, we aimed to provide full information on the state of the ALNS algorithm rather than problem-specific features. In this manner, DR-ALNS does not require changing the state space when applied to a different ALNS for addressing other COPs and shows strong performances when transferred to other problems. We have shown that such MDP formulation enables the learned policy to outperform baselines, even without characteristics of solutions.
>
> We agree with the reviewer that POMDP may be a more suitable formulation, and we will explore advanced DRL algorithms for such a problem setting in the future.
>
> **Why is the numerical reward 5 for an improvement?**
>
> We chose the value of 5  as it reflects the scoring function proposed in the original ALNS. This was found to work well in preliminary experiments.

---

### Official Review · Reviewer_xQhQ · 2024-01-17

**Significance And Importance:** 2
**Soundness:** 3
**Novelty:** 3
**Clarity:** 4
**Overall Evaluation:** 1
**Confidence:** 3

**Weaknesses:**

1: Minor weaknesses that are easily fixable.

**Contributions Of The Paper:**

Deep Reinforcement Learning (DRL). The approach is evaluated on an orienteering
problem with stochastic weights and time windows and also on CVRP, TSP, and
mTSP.

**Ethical Considerations:**

(1) Not Applicable: The paper does not have any ethical considerations to address

**Nomination For Best Paper:**

No

**Questions For Authors:**

My understanding is that ALNS-BO uses a fixed value of parameter dod. I find it
somewhat surprising. In my own implementation of ALNS (for a completely
different class of problems), it pays off to vary the value of dod, even if
choosing it randomly at each step from a predefined set of values. Is it a
common practice to keep the parameter dod fixed? In my mind, DR-ALNS is given
some advantage by allowing the value of dod to fluctuate.

Comparing the running times of DR-ALNS and ALNS-Vanilla/ALNS-BO, it seems that
there is quite a big overhead in DR-ALNS. Is it purely because of the evaluation
of the neural network? Could it be mitigated somehow? For example, what is the
implementation? Is it using GPU for the evaluation? The set of all possible
states is not that big. So, cannot we cache the results on a predefined sample
set of configurations?

I'm somewhat surprised that 3 out of 7 state variables are only about the last
iteration: Best_improved, Current_accepted, and Current_improved. In my
experience, most of the LNS moves are unsuccessful, especially later during the
search. So, it seems to me that the information about the last iteration is not
that important. Or is it? In this regard, it would be nice to have a comparison
similar to Table 4, i.e., test the performance with certain state features
removed.

POST REBUTTAL
==============
Thank you for your answers and additional experiments!

**Reproducibility:**

4: Authors promise to release code and domains (whichever apply).

**Strengths Of The Paper:**

The presented combination of ALNS and DRL is exciting. What I find interesting
is the fact that the neural network doesn't say what to do right now exactly.
Instead, it provides a set of probabilities for choosing a particular
action/configuration within LNS.

Experimental results are good and interesting, especially regarding the training
on one class of problems and testing on another. However, I have some comments
there. See the section Weaknesses.

The paper is well-written and easy to follow.

**Weaknesses Of The Paper:**

In experimental evaluation, the stopping criteria for ALNS variants is a certain
number of iterations (Table 3). I would like much more to see a time limit.
Different variants have different overheads, which are considered
only in Section Computation Time. Because DR-ALNS iteration
takes longer than Vanilla-ALNS (or ALNS-BO), it is not clear if DR-LNS would win
if the stopping criteria were a time limit. This is, in my opinion, the most
important weakness of the paper.

The same issue is in Table 6.

The results for Rise_up and Ratel are copied from the submitted solutions to the
AI4TSP competition. First, it was not clear to me how much time was spent on
those solutions and whether the comparison is fair. Section Computation Time
answers this, but I suggest moving it sooner.

I was a bit confused about Table 5 about *unseen* instances of various sizes. It
initially led me to the (probably wrong) conclusion that the previous results
were made on the instances used for training.

Table 5: It would be good to add also Vanilla-ALNS and ALNS-BO to the table.

As mentioned in the paper, the advantage of DR-ALNS, compared to ALNS-BO, is the
ability to change the weights during the run. In this regard, it would be nice
to demonstrate this ability. For example, by showing how the weights change
during the run when solving one example instance.

Minor comments:
- Table 1: Current_accepted may not be a good name. If the solution was not
  accepted, then it wouldn't be current after all. My understanding is that this
  flag is true if the solution was accepted in the previous iteration.
- A few lines before Section Markov Decision Process Formulation: "search.is".

---

> ### Author Rebuttal · Authors · 2024-01-26
>
> **Time limit stopping criteria**
>
> Incorporating a time-based stopping criterion would indeed provide a comprehensive understanding of the practical applicability. While most of our analysis focused on iteration budget-based performances, we have shown in Fig 3 and Fig 4 that DR-ALNS can find better solutions, even when computation time would be reduced by 80%.
>
> Furthermore, we took the suggestion of the reviewer and performed an extra experiment on CVRP where we ran DR-ALNS and ALNS-Vanilla with a fixed time budget (30 sec). We found that DR-ALNS was 3.6% better than ALNS-vanilla. We will give full results with the same time limit in the final paper.
>
> **Randomly configuring dod**
>
> A fixed 'degree of destruction' (dod) is often used in conventional ALNS. Allowing it to fluctuate could provide an advantage on a case-by-case basis. For example, an advantage is shown by comparing DR-ALNS (OS+ACC) and DR-ALNS (OS+D+ACC) in Table 4.
>
> To further investigate this, we conducted an additional experiment, applying ALNS-vanilla with a randomly chosen dod at each step to solve CVRP. This resulted in solutions that were 3.2% more costly than ALNS-vanilla (with a predefined rate). It means that randomizing dod does not provide an advantage to vanilla ALNS in this case.
>
> **Running times comparison**
>
> The policy inference and the intermediate solution evaluation are performed on CPU. While the evaluation of the neural network contributes to the overhead, the increased runtime is mostly attributed to the greater dod. This necessitates more time for the repair operator due to a greater number of node insertions to conduct.
>
> The stochastic nature of the problem limits the feasibility of caching results, as travel times vary with different random seeds. Despite this, we display in Fig 3 and Fig 4 that DR-ALNS can still find better solutions with much fewer iterations than the ALNS-Vanilla and ALNS-BO. In future, we will try to decrease the overhead of DR-ALNS in each step (e.g., running the neural network on GPU).
>
> **The last iteration state features**
>
> It is plausible that the information from the last iteration holds less significance in the state space. In response to this, we have conducted an additional ablation where we excluded features on the last performed iteration. This resulted in a slight decrease in performance, leading to solutions that are 1.1% more costly. This suggests that the information about the last search iteration holds some importance.

---

### Official Review · Reviewer_qPr5 · 2024-01-24

**Significance And Importance:** 2
**Soundness:** 3
**Novelty:** 2
**Clarity:** 3
**Overall Evaluation:** 1
**Confidence:** 5

**Weaknesses:**

0: Minor weaknesses requiring some work to be addressed for the paper to be accepted.

**Contributions Of The Paper:**

This paper presents a method based on deep reinforcement learning for operator selection and the control of operators and acceptance criteria in the large neighborhood search framework. The proposed approach is applied to instances of the orienteering problem with stochastic weights and time windows (OPSWTW), recently used in a competition. The authors report better results compared to the best methods in this competition and demonstrate that the approach can also be used for other routing problems.

**Ethical Considerations:**

(1) Not Applicable: The paper does not have any ethical considerations to address

**Nomination For Best Paper:**

No

**Questions For Authors:**

How does your approach compare to other hyperheuristic approaches that use reinforcement learning methods?

   Which core ideas used in your algorithm are completely new compared to the literature?

   Which state space features that you use have not been used previously in the literature?

**Reproducibility:**

4: Authors promise to release code and domains (whichever apply).

**Strengths Of The Paper:**

The paper is well-written, and new methods for the dynamic configuration of large neighborhood search are of interest for different problem domains. The proposed approach builds upon previous papers with some small extensions, yet it yields promising results for OPSWTW and can also be applied to other problem domains.

**Weaknesses Of The Paper:**

The contribution regarding the methodology is limited, as it includes several core ideas that have been used previously in the literature. The comparison to the related literature is not complete. Although several approaches based on reinforcement learning are discussed, the discussion of hyperheuristic approaches is too brief (only one recent paper is mentioned). There are also other hyperheuristic approaches that use reinforcement learning and other state features have been introduced in the previous works.

The evaluation can be improved. hyperheuristic approaches are usually evaluated on instances of diverse problems from the CHeSC 2011, the Cross-Domain Heuristic Search Challenge. Some of these problem domains may be interesting for evaluating the current method. It would also be useful to have a more in-depth comparison to Kallestad et al. 2023, as it seems that this paper also considered several other problem domains.

---

> ### Author Rebuttal · Authors · 2024-01-26
>
> **Comparison to other RL-based hyperheuristic**
>
> We compared our method to DRLH (Kallestad, 2023), a DRL application for ALNS (subsection ‘DRLH’). We found DRLH less flexible and effective for several reasons: 1) It uses several computationally expensive operators (e.g., beam search), which is empirically unsuitable for OPSWTW. 2) It adopts destroy operators with predefined destroy degrees. In contrast, our framework selects both destroy operators and their degrees. 3) DRLH uses a fixed rule for SA acceptance, while our method automates the learning of this rule. Besides DRLH, we are aware of Johnn et al. 2023, which is limited to its applicability in routing problems. We do not find other hyperheuristics that use DRL in the context of (A)LNS. We acknowledge the absence of CHeSC2011 in our paper and will cite it with more hyperheuristic papers in our final version. We didn’t compare to hyperheuristics in CHeSC2011 since most of them are not based on ALNS or DRL, which are more suitable for routing problems according to literature. We will attempt to solve CHeSC2011 problem types in the future.
>
> **Novelty in DR-ALNS**
>
> Versatility: A key novelty of DR-ALNS is its problem-agnostic nature, as reflected in the proposed MDP. Unlike many existing methods, DR-ALNS does not require state space or architecture modifications when applied to other ALNS configurations designed for different problems.
>
> Transferability: The problem-agnostic nature enables DR-ALNS to be transferred to different routing problems, even without retraining (Table 7). To the best of our knowledge, none of the existing hyperheuristics approaches that use DRL have demonstrated such transferability.
>
> Comprehensive configuration: DR-ALNS exerts control over the entire configuration of the ALNS framework. This includes the selection of operators, operator parameters and the acceptance criterion. To the best of our knowledge, this work is an early attempt to do this.
>
> **State features**
>
> We introduce novel, problem-agnostic features: cost_difference_best, search_budget, and is_current_best. Cost_difference_best measures the percent deviation from the best cost, and search_budget indicates the utilized percentage of the total budget. This configuration allows these features to be applied across various instance sizes and different problems, as no problem or instance-specific information is included. This enables models trained for a problem to be well transferable to solve other problems (Table 7).

---

### Meta-Review · Area_Chair_KMdh · 2024-02-05

**Recommendation:** Accept (Oral)
**Confidence:** 5

**Metareview:**

We propose to accept the paper for the following reasons:
- a nice combination of ALNS and DRL where DRL is used to select an operator for ALNS;
- applied to the orienteering problem with stochastic weights and time window (from AI4TSP competition) and demonstrated its application to other base routing problems, resulting in a good performance without retraining;
- well-written paper;
- well-completed experimental evaluation;
- outperforms two winning competition methods based on DRL;
- points raised by reviewers were well responded to in the rebuttal, and the authors even promised to extend experiments as suggested in one of the reviews.

We ask the authors to reflect the comments of the reviewers as proposed in the rebuttal. In addition, please take care of the following comments:
1. It is important to discuss existing works considering problem-agnostic state features. More specifically, in the context of the hyperheuristics mentioned in reviews, problem-agnostic state features were already considered in "Kletzander, L., Musliu, N., Large-State Reinforcement Learning for Hyper-Heuristics. AAAI 2023." so further discussion on the novelty of the proposed features is necessary in the final version of the paper.
2. Authors mentioned in their rebuttal, 'We compared our method to DRLH (Kallestad, 2023), a DRL application for ALNS (subsection ‘DRLH’).' While this comparison was made, it appears that Kallestad et al. 2023 considered several other problem domains as well. The authors should reflect in the final version whether and how their approach works on other problem domains.

Looking at the paper from a future perspective, the meta-reviewer recommends consideration of other types of problems. Extending given orienteering problems with capacitated vehicles or salesmen introduces a rather simple extension. How would the approach behave on more complex problems where more features are introduced?

**Ethical Considerations:**

(1) Not Applicable: The paper does not have any ethical considerations to address